# Biolinguistics: A Scientometric Analysis of Research on (Children’s) Molecular Genetics of Speech and Language (Disorders)

**DOI:** 10.3390/children9091300

**Published:** 2022-08-27

**Authors:** Ahmed Alduais, Shrouq Almaghlouth, Hind Alfadda, Fawaz Qasem

**Affiliations:** 1Department of Human Sciences, University of Verona, 37129 Verona, Italy; 2Department of English, King Faisal University, Al-Ahsa 31982, Saudi Arabia; 3Department of Curriculum and Instruction, King Saud University, Riyadh 11362, Saudi Arabia; 4Department of English, University of Bisha, Al-Namas 67714, Saudi Arabia

**Keywords:** biolinguistics, language gene, language faculty, nature–nurture dichotomy, FOXP2, language biological bases, evo-devo, scientometric review

## Abstract

There are numerous children and adolescents throughout the world who are either diagnosed with speech and language disorders or manifest any of them as a result of another disorder. Meanwhile, since the emergence of language as an innate capability, the question of whether it constitutes a behaviour or an innate ability has been debated for decades. There have been several theories developed that support and demonstrate the biological foundations of human language. Molecular evidence of the biological basis of language came from the FOXP2 gene, also known as the language gene. Taking a closer look at both human language and biology, biolinguistics is at the core of these inquiries—attempting to understand the aetiologies of the genetics of speech and language disorders in children and adolescents. This paper presents empirical evidence based on both scientometrics and bibliometrics. We collected data between 1935 and 2022 from Scopus, WOS, and Lens. A total of 1570 documents were analysed from Scopus, 1440 from the WOS, and 5275 from Lens. Bibliometric analysis was performed using Excel based on generated reports from these three databases. CiteSpace 5.8.R3 and VOSviewer 1.6.18 were used to conduct the scientometric analysis. Eight bibliometric and eight scientometric indicators were used to measure the development of the field of biolinguistics, including but not limited to the production size of knowledge, the most examined topics, and the most frequent concepts and variables. A major finding of our study is identifying the most examined topics in the genetics of speech and language disorders. These included: gestural communication, structural design, cultural evolution, neural network, language tools, human language faculty, evolutionary biology, molecular biology, and theoretical perspective on language evolution.

## 1. Introduction

### 1.1. The Rise of Biolinguistics

The study of language has been always a pivotal corner in human history; and linguistic affiliations with other disciplines are—not surprisingly—profound and revolving continuously. Perhaps one of the earliest of such affiliations is the emergence of biolinguistics, which has ancient roots in human philosophy [1] (p. 926). As suggested by its name, biolinguistics promotes a program for the study of the biology of language [2]; thus, infusing together elements from linguistic enterprise and biology as a natural science. This could be traced back all the way to the days of Aristotle who expressed relevant philosophical underpinnings with regards to the similarities the systems of human and animals’ communication, birdsongs, in particular, have in common [3,4]. Fast forward to the seventeenth century and the birth of modern science and natural philosophy with Galileo and Descartes, this affiliation can be detected even stronger. However, it was not until the year 1950 that the term biolinguistics was mentioned for the first time in a work by Meader and Muyskens [5].

Such mention, however, was passed unnoticed with second to non-referencing to this work. What was noticed at that time, however, is the production of a number of landmark works that signalled the official birth of biolinguistics as a scientific inquiry. Piattelli-Palmarini [6] (p. 13) refers to this period as the “early days” of biolinguistics. The first of these landmark works are Noam Chomsky’s “Logical Structure of Linguistic Theory” [7] and “Syntactic Structure” [8], along with his famous critique [9] of behaviourism and Skinner’s “Verbal Behaviour” [10]. Such review has documented a strong shift in language studies from structural linguistics to a biological take on language study, which was later translated in the advent of Chomsky’s generative grammar in “Aspects of the Theory of Syntax” [11]. Within such conception, the Language Acquisition Device (henceforth, LAD) has been proposed to account for the acquisition/ evolution of language. Universal grammar (UG) is another fundamental cornerstone within generative grammar in which the premise is that UG allows for the genetic endowment of the language faculty [12] (p. 1) “understood to be a cognitive organ”. To illustrate, UG presupposes the innate presence of a limited set of constraints in charge of organizing human language, any human language and regardless of its external linguistic manifestation; and that such innateness is shared by all humans as a “species property” unique to them. Chomsky’s linguistic contributions were paralleled by the work of the biologist Eric Lenneberge’s, “Biological Foundation of Language” [13]. All this was motivated by the rejection of the potential rule of communication in constructing language evolution as well as the increased interest in poverty of stimulus and innateness in language studies [6].

This affiliation, however, was not confirmed and coined as biolinguistics until 1974, when the latter publicly used it while organizing a conference along with Chomsky and the biologist Salva Luria in Massachusetts. As a promising interdisciplinary enterprise, many conferences, research groups and works were inspired by this new affiliation at that time up until the mid-1980s, marking what Piattelli-Palmarini [6] refers to as the middle period of biolinguistics. In the years between 1985 and 2000, works on linguistic theory have maturated further and begun to appear more comprehensive theoretically with the advent of the “Theory of Government and Binding” [14] (p. 15). Consequently, this evolution reached another landmark within biolinguistics, the emergence of the minimalist program [15]. The minimalist program has been set forth as potentially an interdisciplinary take on generative grammar that was both a “revision of Chomsky [early] linguistics” as well as an “extension” and “reconstruction” of previous literature within biolinguistics [16] (p. 1). Chomsky [17] links his ideas on biolinguistics to the older philosophical literature on language evolution, acquisition and usage, respectively referring to these as Humboldt’s problem, Plato’s problem and Descartes’ problem. In that sense, it is not surprising that previous work on biolinguistics, since the mid-1950s to that time, was rather considered a “cognitive revolution” [18] (p. 1). Keeping in mind that words such as “cognitive science” were considered new “buzzwords” [18] in academia during that period, this revolution began to attract more interest across different disciplines. Such interdisciplinary nature marked the consolidation of biolinguistics from merely an “initial program” to a “full domain” [6] (p. 15). It also, the latter adds, marked the emergence of the third period in biolinguistics which continues until the present time. Lyle Jerkins [18] edited a comprehensive book by the turn of the millennium which was quite representative of the more recent trends of biolinguistics and its overarching interdisciplinary premises.

Since its early days, biolinguistics has at its heart the desire to incorporate biolinguistics within natural sciences [18] (p. 5), which highlighted the “unification” problem in parallel lines to this goal. Originally, unification has its roots in physics and refers to the “synthesis of subfields”, thus linking diverse objects together [19] (p. 317). Going along the same line, Mendívil-Giró [20] (p. 21) subscribes to the language uniformation hypothesis; rejecting linguistic relativity and emphasizing that there are no primitive languages as they all exhibit similar evolution and development. As an interdisciplinary domain, biolinguistics draws a link between Chomsky’s universal grammar and Goethe’s theory of Urform [21]; both of which emphasized the presence of a universal constant in all underlying structures. This was clearly exhibited with the publication of another landmark work, a joint article by Marc Hauser, Noam Chomsky and Tecumseh Fitch entitled “The Faculty of Language: What is it, who has it, and How did it Evolve?” in Science, [22]. Within this perception, the recursive property of language is intensified, and a distinction is made between faculty of language broadly defined (FLB) and faculty of language narrowly defined (FLN). In FLN, the focus is on aspects of language that are peculiar to language while in FLB it encompasses other parts shared with other psychological capacities.

This period also revealed further work on the minimalist program and language faculty. Chomsky [12] (p. 1) highlighted the I language (internal language) a state which results from mind/brain computational systems in order to generate “structured expressions”. Such expressions are based on instructions motivated by two sub-systems working collaboratively; the conceptual intentional interface and the sensory–motor interface, which externalizes such expression in production. These two sub systems are heavily linked to Merge; a single core computational mechanism going along the lines of unification [12,23]. As it stands, Merge minimalized generative grammar to one mechanism operating language faculty. It is also possible to see Merge as a revision, which further questions the dichotomy between FLB and FLN [3].

All these advances in the literature that were cultivated during the twenty-first century produced a sincere renowned interest in biolinguistics; thus, endowing this period with a state of “renaissance” of biolinguistics [24] (p. 3). The same authors have been motivated to call it as such due to the brief popularity the term biolinguistics has in the mid-seventies, which did not last well until the turn of the century [24]. Piattelli-Palmarini (p. 13), acknowledges the fact the progress and advancement in biolinguistics were, despite being “fascinating”, rather “slow” [6]. A multitude of factors could be credited as motivating such a renaissance. Perhaps the first of these is the new appreciation and celebration in academic work of multidisciplinary and interdisciplinary endeavours incorporating language studies with other natural sciences. Such appreciation has been translated explicitly in Jenkins’ aforementioned edited volume [18]; many of its contributions have been reviewed as comprehensive introductory texts [25]. Similarly, Sciullo and Boeckx, edited a comprehensive volume in biolinguistics, which was perceived as the result of two interdisciplinary conferences in 2007 [26].

This interdisciplinarity was accompanied by a shift in perspective in a number of relevant fields, Boecks [24], for instance, states such transformation within works in comparative psychology, along with an extensive expansion of scope in biology towards more pluralist and internalist views. Elaborative research on the forkhead box P2 gene (FOXP2), for example, with its strong connections to the genetic endowment of language has been motivating such a renaissance as well. However, it was the rise of the evolutionary developmental (henceforth, evo-devo) framework [27] in biology that has a more prominent role in expanding the plethora of relevant biolinguistics works. Hence, it was not surprising that the term biolinguistics became a buzzword itself in linguistic inquiry [28] (p. 926).

Consequently, all these efforts resulted in the launch of a new specialized peer-reviewed open-access journal, Biolinguistics in 2007 by the University of Cyprus and it has issued fifteen annual issues so far along with two special issues in 2020 and 2017. While this is not the first attempt to launch a specialized journal in biolinguistics, Jenkins attempted to do so and actually managed to receive documented support earlier but such a journal never materialized, “Biolinguistics” as a specialized journal was needed in response to the increasing renewed interest in generative grammar [29]. Such interest, along with the aforementioned appreciation of interdisciplinarity in science, has motivated experienced scholars from all over the world to continue exploring biolinguistics investigations despite facing some theoretical and computational challenges impeding their work [30]. Boeckx, [31] (p 316), states that biolinguistics, as a “cognitive science”, tackles issues such as the genome and the cognitive profile of the human species, that we might perhaps know “little” of; nevertheless, he adds that our current state of knowledge now has improved a lot to face such tasks. Since 2020, Biolinguistics has been indexed in Scopus and Web of Science core collection (Emerging Sources Citation Index, ESCI). In March 2022, however, the journal migrated to another publishing platform (https://bioling.psychopen.eu/) and under the publisher PsychOpen.

### 1.2. The Scope of Biolinguistics

The previous discussion highlighted the rise and emergence of biolinguistics from a mere “dream” [6] to a full-blown field of inquiry with diverse applications. Attempting to draw a defining map of the scope of biolinguistics can be somehow challenging. This is primarily because one can differentiate in the relevant literature between two views of biolinguistics. Martins and Boecks [32] (p. 1) identify the first as relating to works which are inclined towards generative persuasion and theoretical linguistics while the second rather departs from linguistic inquiry towards a more biology-based orientation. In that sense, it is possible to link such perceptions to what Boecks and Grohmann [33] refer to as weak and strong versions of biolinguistics. Clearly, more linguistic-based works following Chomsky’s generative grammar are on the weaker side; while works taking insight from more evolutionary and biology-based inspirations such as Lenneberg’s are rather stronger. In fact, it is within the stronger version that Boeckx and Benitez-Burraco [34] (p. 3) highlight the emergence of biolinguistics [34] which departs from associations with minimalism and generative grammar towards the adoption of recent biological programs such as evo-devo. Consequently, Pleyer and Hartmann [35] (p. 14) call for a more inclusive approach of “progressive biolinguistics” that would converge occasionally with usage-based approaches to linguistic investigation instead of contradicting them.

In accordance with this, it is possible to identify many fields of study as included within biolinguistics. Di Sciullo and Jenkins [25] (p. 277) enumerate an extensive list of what subfields constitute biolinguistics including:Theoretical linguistics (syntax, morphology, phonology, lexicon, etc.)Computational linguistics and parsingMathematical modelling and simulationChild language acquisition and multilingualismComparative linguistics (e.g., typology)Perceptual studiesSpeech disorders (developmental verbal dyspraxia, language impairment, dyslexia, etc.)Cross-species comparative work (nonhuman communication, songbirds, etc.)Language changeLanguage contact (pidgins, creoles, etc.)

However, due to such broad and ever-expanding scope of investigation, biolinguistics has been faced with some criticism. For example, Behme [36] criticizes the foundation of biolinguistics for being “inherently incoherent”. Going along the same line of thought, Bickerton [37] casts the minimalist approach as being ambiguous besides highlighting some biological underpinnings in biolinguistics works as either misunderstood or overestimated. Martins and Boeckx [32] criticize current work affiliated with biolinguistics for failing to address linguistics and biology appropriately by ignoring biological inspirations extensively and not investing in sufficient linguistic theorization. By the same token, Bowling [38] highlights the misleading nature–nurture dichotomy that perpetuates biolinguistics literature. The next section highlights some recent works within biolinguistics that attempt to bridge such gaps.

### 1.3. Scientific Contributions for Biolinguistics

Starting with the last point, the nature–nurture dichotomy, Kirby [39] attempted to tackle this by emphasizing the role of cultural transmission in language evolution. Going along the same line, Pleyer and Hartmann [35] emphasize some areas of convergence in recent years between biolinguistics and usage-based language approaches such as innateness, cultural and biological evolution as well as domain specificity and modularity. In their biolinguistics investigations, Balari et al. [40] (p. 489) highlight their interest in the “fossils of language” despite the complexity of such a task, given the complete absence of any consensus in this regard. Again, such work demonstrates the rather biological imputes that continues to grow with recent literature. Going along the same lines, de Aquino Silva and de Motta Sampaio, [41] as well as Mao [42], reveal similar interest as they attempt to present a primarily biologically oriented investigation to formulate the evolutionary map of human languages and language faculty. In another work by Bolender [43], however, biolinguistics inspiration is used in conjunction with calculi as an attempt to consolidate the link between biolinguistics and natural sciences through Merge and language recursive operation. Other works can be classified as primarily linguistic based, falling within the subfield of syntax; for instance, Trettenbrein’s [44] work on UG as an axiom and not a hypothesis or the work of Brody’s [45] on one-dimensional syntax. This is beside works investigating language acquisition such as Feeney’s [46] work on dual-processing and Rahul and Ponniah’s [47] work on incidental vocabulary learning. Evidently, this is only a small fraction of recent studies from the rich and interdisciplinary biolinguistics literature.

All in all, it is possible to see in this concise review that biolinguistics, as an interdisciplinary field of inquiry, has revolved and blossomed tremendously over the course of the last seventy years. Being described across different times as a quite promising field of inquiry [25,48], it is possible to say that biolinguistics has continuously had high expectations as it unfolds. Fitch [49] (p. 455), suggests that with the way this “broad, data-driven” field unfolded throughout these decades, it is fair to say that it has “aged well”.

### 1.4. Molecular Genetics of Speech and Language (Disorders)

Genetic studies of speech and language have established a new trend in the study of the biological bases of human language [50]. While much effort has been expended to identify molecular aspects of the human language faculty through human genome analysis [51], to date, only a few genes have been identified as contributing to the genetics of human language. These include FOXP2 [52] (i.e., oral motor sequencing abilities [53]), microcephalin (MCPH1) (i.e., language delay [54,55]), Contactin-associated protein-like 2 (CNTNAP2) [56] (i.e., language processing [57]), and abnormal spindle-like microcephaly (ASPM) [50] (associated to lexical tone perception [54,58]). There is no doubt that the study of genetic disorders of speech and language is an important contribution to our understanding of the biological bases of language [59]. As a result of this molecular approach, it is possible to disseminate knowledge about the neurological pathways responsible for speech and language impairments [60]. Most of the research, however, has been focused on developmental disorders manifesting in speech and language [61] (e.g., dyslexia [62]).

The FOXP2 gene, also known as the language gene, has been implicated in speech and language disorders based on imaging techniques and mice-mutated data [63]. This gene was first introduced in 2001, making the first attempt to study the molecular genetics of language and speech [64]. The forkhead domain gene was found to be mutated in a severe speech and language disorder, and FOXP2 was found to be involved in the development of speech and language [65]. Researchers have, however, demonstrated that this gene may not be applicable to all types of disorders [66] and it has not been confirmed that it is involved in autism or specific language disorders [67,68,69]. However, a study conducted on the Chinese population claimed that FOXP2 played a significant role in the pathogenesis of autism [70]. Even though there is considerable evidence that autism is an inheritable condition, it remains controversial [71].

Several types of speech and language disorders overlap, which is another challenge for researchers studying the genetics of speech and language. According to a study that examined speech sound disorder, language impairment, and reading disability, they remain distinct concerning comorbidity subtypes [72]. Accordingly, several types of disorders are inherited, but the identification of their molecular aetiologies remains a questionable aspect although FOXP2 has contributed somewhat to probing this ambiguous human aspect [73]. A linguistic, neurolinguistic, and cognitive science interdisciplinary perspective is necessary in order to better understand the biological bases of human language, according to Grimaldi [74]. Similarly, another study recommended integrating language sciences, genetics, neurobiology, psychology, and linguistics in order to gain a better understanding of human language faculty from a biolinguistics viewpoint [75].

Recent studies indicate that the study of molecular genetics of speech and language remains complex [76] regardless of the exponential growth of research in this area [77]. In addition, when considering the nature–nurture aspects of language, although theories of language such as universal grammar have contributed to our understanding of the innate aspects of language, the environment still plays a significant role in language acquisition and learning [78]. As the study of molecular genetics of speech and language expands, new concepts are being introduced, such as the faculty of language broad sense, faculty of language derived components, and faculty of language narrow sense [53]. In conclusion, two recent reviews summarized evidence on the importance of studying molecular brain aspects in understanding neurodevelopmental disorders [79,80].

### 1.5. Purpose of the Present Study

Numerous studies have examined the field of biolinguistics from a variety of perspectives and with a variety of foci. One study examined biolinguistics in the context of presenting mathematical models to explain the evolution of human language via natural selection [81]. One study assessed critically the emergence of biolinguistics and biosemiotics (i.e., distinctions between nature and culture) as two similar disciplines but with points of difference to differentiate them from one another [82]. Another paper reviewing cultural evolution and genetic evolution of language concluded that more evidence supports cultural evolution than genetic evolution [83].

In connection with this study is the study reviewing the evidence regarding FOXP2 in relation to the identification and evaluation of genetic evidence for language disorders. The FOXP2 gene was found to be essential for typical language and speech development [84]. Similar research reviewing the biological basis of language through typical and atypical language development focused on specific language impairment [85]. An additional review of the emergence of biolinguistics concluded that it combines information from multiple fields, including genetics, neurology, neuroscience, psychology, linguistics, and evolutionary biology [86]. Wu attempted to review biolinguistics from a number of points of view. This included literature statistics related to biolinguistics, proceedings and conference papers, book reviews, and a survey of biolinguistics proponents [87].

The field of biolinguistics has not yet been examined using bibliometric and scientometric measures to map its knowledge domains. This paper sought to assess the scientific contributions of biolinguistics by quantifying the volume of knowledge that has been produced and the key contributors (i.e., authors, countries, universities and journals). It focuses on the current and foreseeable directions of biolinguistics, including how it will be incorporated into other disciplines between 1935 and 2022. Thus, we raised three main questions to guide conducting this study. (1) What is the knowledge production size of biolinguistics research measured by year, region, higher education institution, journal, publisher, research area, and author? (2) What are the most explored themes and examined topics in biolinguistics? (3) Who are the central authors establishing for a better understanding of biolinguistics and who are those receiving greater attention from researchers in the field?

## 2. Methods

### 2.1. Research Methods

Scientometrics pertains to the “study of artifacts; one examines not science and scholarship but the products of those activities” [88] (p. 491). In scientometrics, researchers examine “the quantitative aspects of the production, dissemination and use of scientific information with the aim of achieving a better understanding of the mechanisms of scientific research as a social activity” [89] (p. 6). There is some debate regarding whether this type of research is intended to assess the quality of published knowledge or not. Previous research indicated that “the task of determining quality papers is especially difficult in BIS [bibliometrics, informetrics and scientometrics] due to the very heterogeneous origin of the researchers” [90] (p. 390). Irrespective of this controversy, the basic goals of such studies are to “reveal characteristics of scientometric phenomena and processes in scientific research for more efficient management of science” [91] (p. 1).

As part of scientometric studies, scientometric indicators serve to guide the design and analysis of the study. These include elements (e.g., publication, citation and reference, potential, etc.) or type indicators (e.g., quantitative, impact) [91]. A frequent concept used while conducting such studies is ‘mapping knowledge domains’ which refers to making “an image that shows the development process and the structural relationship of scientific knowledge”—using maps that are “useful tools for tracking the frontiers of science and technology, facilitating knowledge management, and assisting scientific and technological decision-making” [92] (p. 6201). Nowadays, this research is becoming more expanded to include all areas of study, rather than remaining confined to purely medical and health sciences [93]. The present study explored biolinguistics as a subfield of linguistics, which enables integration with other fields, such as biology and neuroscience.

### 2.2. Measures

Bibliometric and scientometric indicators are both considered tools to guide the assessment of knowledge produced in a particular field (e.g., biolinguistics) [94]. Bibliometric indicators are often provided in knowledge databases (e.g., Scopus, WOS, and Lens) [95,96,97,98]. Scientometric indicators are usually provided through scientometric software. For instance, in this SmR, we used CiteSpace 5.8.R3 [99] and VOSviewer 1.6.18 [100]. Table 1 summarises the bibliometric and scientometric indicators used in this study.

### 2.3. Data-Collection and Sample

Data were retrieved from three databases: Scopus, WOS, and Lens. They were included for a number of reasons. A first limitation of Scopus and WOS is that they are limited to publications that include their indexed journals and other publications [95,96,97]. Additionally, Lens contains more data than either Scopus or WOS [98].

Searches were conducted on Tuesday, 18 April 2022. The language limitations were not imposed if titles, abstracts, and keywords were provided in English. Because few results were available in other languages, manual verification was conducted. Articles, review articles, book chapters, books, conference proceedings (full papers), dissertations, as well as early access publications of these types, were considered. Table 2 contains the search strings for the three databases and other specifications.

This study examined the use of the concept of “biolinguistics” and any comparable concepts to quantify the development and size of research in this field. As a result, we included additional keywords in order to broaden the search results. Among these were, for example, “biology of language” and “language evolution”. An initial search on Google and previous knowledge of the field indicated using the above search strings when searching for information related to biolinguistics.

### 2.4. Data Analysis

A number of steps were taken before the data were analysed. Initially, the Scopus data were exported to three different formats: Excel sheets for the bibliometric analysis, RIS files for CiteSpace, and CSV files for VOSviewer. According to CiteSpace’s requirements, the RIS file was converted to WOS. Furthermore, WOS data were extracted in two formats: as text files that were converted to Excel sheets for bibliometric analyses, and as plain text files for CiteSpace and VOSviewer. In conclusion, Lens data were extracted in two formats: CSV for bibliometric analysis, and full-record CSV for VOSviewer.

CiteSpace and Mendeley were used to remove duplicate documents prior to CiteSpace analysis. Excel was used to perform the bibliometric analysis. We generated the tables for the citation reports using Microsoft Excel and converted them into figures.

Default settings for scientometric analysis were set for both software packages. The three databases were analysed separately, including network visualizations, overlay visualizations, and density visualizations. The analyses were conducted three times for Scoups and WOS: cooccurrence analysis by author keyword, co-citation analysis by source, and co-citation analysis by cited author. Lens underwent four analyses: cooccurrence analysis by keyword, citation analysis by author, citation analysis by source, and citation analysis by document. As a result of our analysis of CiteSpace for Scopus and WOS, we have obtained the following information: co-citations by document (references), co-citations by cited authors, and occurrences (keywords). Summary tables, cluster summaries, visual maps, and burst tables were used to summarize the findings.

We should highlight at the conclusion that although we attempted to combine the data from the three databases into a single result, we encountered various technological difficulties. First, each of the employed software packages is configured to independently analyse the data from each of these databases. In other words, it needs to convert all the data from the three databases into a single format, which was not achievable on our end. Second, we wanted to determine if there are substantial variations between the three databases in terms of the bibliometric and scientometric indicators used. We wanted to emphasize the importance of using several databases for these types of investigations, but we also recommend integrating the data during data analysis wherever possible.

## 3. Results

### 3.1. Result Overview

Our findings were split into two categories. First, we provided bibliometric biolinguistics indicators. Data from Scopus, WOS, and Lens databases were used to create the indicators. The top 10 countries, universities, journals, publishers, subject/research areas, and authors are just a few examples of bibliometric indicators. The second section of the paper presents scientometric indicators for the growth of biolinguistics. These indicators were analysed using VOSviewer and CiteSpace. The analysis included indicators such as citation, co-citation, and cooccurrence.

In the first subsection, several bibliometric indications for the evolution of biolinguistics were offered. These included the number and types of publications, the volume of biolinguistics’ knowledge output by year, region, university and/or research centre, journal, publisher, research area, keywords and cooccurrence, and author. In the second section, we provided visual representations and tabular representations of the scientometric indicators used to measure the growth of biolinguistics. Included were the top keywords with the strongest citation bursts, the top keywords with cited authors and clusters, the cooccurrence of keywords used by authors, (co)-citation by author, (co)-citation by source, and the most cited papers in Scopus, WOS, and Lens. In addition, we employed additional scientometric indicators to emphasise the impact of research on biolinguistics by identifying the most important and central authors, as well as those whose citations have the potential to increase. 

### 3.2. Bibliometric Indicators for the Study of Biolinguistics

#### 3.2.1. Overview of Biolinguistics Studies from Scopus, Web of Science, and Lens

For analysis, 1570 Scopus documents, 1440 WOS documents, and 5275 Lens documents on biolinguistics were retrieved. Moreover, 1973–2022, 1988–2022, and 1935–2022 were the data periods for the three databases. There were 961 articles, 181 review articles, 159 book chapters, 41 books, and 228 conference papers among the Scopus documents. The WOS produced 975 articles, 122 review articles, 69 (book) chapters, 8 early access papers, and 202 proceeding papers. There were 3420 articles, 614 unknown types, 423 book chapters, 276 books, 55 dissertations, and 351 conference proceedings (article) and preprints among the Lens documents. The majority of these documents were written in English, with others in Spanish, Russian, French, Portuguese, German, Italian, Chinese, etc. Since the analysis was based on the title, keywords, abstract, and references, all of these elements were included in the English language. This inclusion was considered to prevent bias towards English-language publications.

#### 3.2.2. Biolinguistics Knowledge Production Size by Year 

Figure 1A–C shows the length of production by year for the three databases. As can be seen, there has been a significant rise in knowledge production in biolinguistics reaching its peak in 2018 in Scopus with 139 publications, 2018 in the WOS with 136 publications, and 2016 in Lens with 435 publications. The range of publications per year is 1–139 in Scopus, 1–136 in the WOS, and 1–435 in Lens. The lowest number of publications occurs in previous years in all databases. In addition, of the 8285 biolinguistics publications published between 1935 and 2022, 7797 were published between 2000 and 2022. This means, there has been a rise in the production of knowledge related to biolinguistics in the last two decades.

### 3.3. Production of Biolinguistics Research by Country and University

Figure 2A–C shows the top 10 producing countries for knowledge related to biolinguistics. The US ranks first and the UK ranks second in all databases. The rest of the 10 ten countries producing knowledge in biolinguistics are all European except Australia, Japan, and China.

Figure 3A–C presents the top 10 universities and/or research centres producing knowledge in biolinguistics. As is seen, the UK has the top institutions producing knowledge in biolinguistics, namely, the University of Edinburgh, followed by the Max Planck Society in Germany and the Max Planck Institute for Psycholinguistics in the Netherlands. The League of European Research Universities located in Belgium ranks first in the WOS database.

### 3.4. Production of Biolinguistics Research by Journal and Publisher

Figure 4A–D demonstrates the top 10 journals publishing research in biolinguistics. The journals vary between several disciplines including psychology, cognitive science, biology and language studies. The top journal is Frontiers in Psychology. Interestingly, we can see a journal titled ‘Biolinguistics’ listed on three databases but appears among the top 10 only on Lens. Figure 4D shows an extended list of journals based on publishers. On this list, we can see that most of these journals are related to biological sciences, neurosciences with a few journals in psychology and linguistics.

Figure 5A,B shows the list of top 10 publishers for knowledge in biolinguistics. These lists are limited to the WOS and Lens databases as Scopus does not include publisher information. It can be seen that “Elsevier” and “Springer Nature” are the top two publishers for sources publishing knowledge in the field of biolinguistics. Frontiers Media also plays a vital role in publishing literature related to biolinguistics.

### 3.5. Production of Biolinguistics by Research Area, Keywords, and Cooccurrence

Biolinguistics is a field of study in linguistics which integrates mainly with biology and other fields as shown in (Figure 6A–C). Figure 6A indicates that the top four subject areas publishing in biolinguistics are social sciences, arts and humanities, psychology, and computer science. Figure 6B shows that linguistics, psychology, computer science, and neurosciences are the top four research areas relating to biolinguistics. These are further confirmed in Figure 6C where computer science, linguistics, psychology, and language evolution are introduced as the top four fields of study publishing in biolinguistics. Lens shows more specific fields that are related to this field of study (e.g., language evolution, natural language processing, and language acquisition).

### 3.6. Production of Biolinguistics by Authors

Contribution to biolinguistics is neither measured by quantity nor by quality albeit these are two indicators of influential works and/or authors in the field. However, we intended to show the authors who produced more knowledge related to biolinguistics as shown in (Figure 7A–C). As is seen, Benitez-Burraco [104], Kirby [105], Christiansen [106], and Boeckx [31] are among the top contributors in the field.

### 3.7. Scientometric Indicators for the Study of Biolinguistics

Overview of Biolinguistics Studies from Scopus, Web of Science, and Lens

This section presents the scientometric analysis for the retrieved data from Scopus, WOS, and Lens databases. It focusses on highlighting the impact of certain concepts, authors, references, and emerging trends on the field of biolinguistics.

We first showed the top keywords with the strongest citation bursts using CiteSpace for data from Scopus and WOS (Figure 8A,B). The green line indicates the period for all research. The red line indicates the beginning and end of the burst period. The word with the strongest citation burst in Scopus is (human experiment = 11.36) between 2019 and 2022, and (cultural revolution = 8.51) between 2017 and 2020 for the WOS. The citation burst changes according to the database. For instance, we can see biological evolution, formal language, etc., in Scopus only but gene, emergence, language faculty, etc., in the WOS.

These are further illustrated with clusters and authors in network visualisations (Figure 9A–D). Figure 8A shows topics such as multilevel selection, new-born monkey, among others, as the most explored topics in biolinguistics. More specific concepts are shown in Figure 9B and these include iterated learning, language development and Bantu language. Figure 9C,D show the most cited authors and the topics being searched while citing these authors. These topics include gestural communication, biolinguistics, etc. (see Figure 9C). In the WOS database, they include other words such as human language, language, etc. (see Figure 9D). The key to comprehending the logic of these visual maps is based on the intensity of the text and lines listed next to each cluster. For instance, the cluster containing the number 0 for gestural repertoire size is the best cluster because it contains the most authors and keywords related to this cluster. Similar logic could be applied to the remaining clusters. The clusters are ranked from 0 to 12 according to the amount of research conducted on each cluster, which corresponds to the intensity of the text next to each cluster. This is applicable to the remaining figures.

Another important factor is the cooccurrence of used keywords. Using VOSviewer, we generated three visual network maps for the occurrence of the most used keywords in biolinguistics in the three databases (Figure 10A–C). Each colour represents one direction for the study of biolinguistics. For instance, green shows topics related to cognitive historical linguistics, blue to gesture and vocal learning multimodal (see Figure 10A). These colours change according to the database. For instance, in Figure 10B, green indicates language evolution, blue for biolinguistics, and purple for natural selection and phylogenetics. Orange in Figure 10C shows keywords related to biolinguistics.

Using VOSviewer, we generated three visual network maps for co-citation and citation by author (Figure 11A–C). Each colour represents a network for the co-citation or citation for authors. The larger the size of the circle, the more co-cited or cited is the author. We can see similar author repeated in the three databases be it for co-citation or citation. Among these are Kirby [105], Chomsky [12], Pinker [107], and Arbib [108].

Using VOSviewer, we generated three visual network maps for co-citation and citation by source (Figure 12A–C). Each colour represents a network for the co-citation or citation for sources. The larger the size of the circle, the more co-cited or cited is the source. For instance, in Figure 12A, journals in red are more related to language studies, journals in blue are more related to neuroscience, and journals in green are more related to psychology and biological sciences. These journals seem to be similar in Figure 12B using the WOS database. Figure 12C shows the citation network for journals in the Lens database. Among these are Frontiers in Psychology, The Evolution of Language, etc.

Using the bibliometric data provided in Scopus, WOS, and Lens, we exported the citation reports and reported the top 10 cited works (Table 3). Based on the database, it is evident that the top cited sources vary. After merging the top 10 sources from each database, 20 sources are provided instead of 30. Particularly, the sources listed in Lens differ from those in Scopus and the WOS, and this may be due to their restricted inclusion criteria. Furthermore, the sources from Lens contained a greater number of citations. As an example, Scopus’ number two citation in biolinguistics has only 949 citations as compared to Lens’ 1825. In all fairness, it can be observed that all of these top-cited works have some connection to biolinguistics.

### 3.8. Impact of Research on Biolinguistics by Clusters, Citation Counts, Citation Bursts, Centrality, and Sigma

#### 3.8.1. Clusters

The network is divided into 21 co-citation clusters in Scopus data (Table 4). The largest 8 clusters are summarised as follows. The largest cluster (#0) has 210 members and a silhouette value of 0.728. It is labelled as gestural communication by LLR, language evolution by LSI, and year (1.53) by MI. The most relevant citer to the cluster is “Creating Language: Integrating Evolution, Acquisition, and Processing” [128].

The network is divided into 14 co-citation clusters in the WOS data. The largest 5 clusters are summarized as follows. The largest cluster (#0) has 171 members and a silhouette value of 0.791. It is labelled as exorcising Grice’s ghost by LLR, language evolution by LSI, and role (0.92) by MI. The most relevant citer to the cluster is “Empirical approaches to the study of language evolution” [129] (Table 4).

#### 3.8.2. Citation Counts

In Scopus, the top-ranked item by citation counts is Chomsky [130] in Cluster #4, with citation counts of 378. The second one is Tomasello [131] in Cluster #0, with citation counts of 340. In the WOS, the top-ranked item by citation counts is Hauser [132] in Cluster #1, with citation counts of 316. The second one is Chomsky [133] in Cluster #1, with citation counts of 308 (See Table 5).

#### 3.8.3. Bursts

In Scopus, the top-ranked item by bursts is Batali [145] in Cluster #3, with bursts of 20.38. The second one is Steels [140] in Cluster #3, with bursts of 16.56. In the WOS, the top-ranked item by bursts is Steels [113] in Cluster #3, with bursts of 19.03. The second one is Batali [146] in Cluster #3, with bursts of 17.45 (see Table 6). These are further demonstrated in Figure 13A–D.

#### 3.8.4. Centrality

In Scopus, the top-ranked item by centrality is Donald [158] in Cluster #0, with centrality of 108. The second one is Bates [159] in Cluster #0, with a centrality of 103. In the WOS, the top-ranked item by centrality is Kirby [105] in Cluster #2, with a centrality of 103. The second one is Bickerton [37] in Cluster #1, with centrality of 99 (see Table 7).

#### 3.8.5. Sigma

In Scopus, the top-ranked item by sigma is Donald [158] in Cluster #0, with a sigma of 0.00. The second one is Bates [159] in Cluster #0, with a sigma of 0.00. In the WOS, the top-ranked item by sigma is Kirby [105] in Cluster #2, with a sigma of 0.00. The second one is Bickerton [37] in Cluster #1, with a sigma of 0.00 (see Table 8).

## 4. Discussion

This study intended to identify the scientific achievements of biolinguistics by analysing the volume of knowledge created and the contributions of notable researchers (i.e., authors, countries, universities, and journals). It examined the current and future directions of biolinguistics, as well as its integration with and relationship to other disciplines. The study featured two primary indicators, bibliometric indicators acquired from the Scopus, WOS, and Lens databases, which included publications by year, the top 10 nations, universities, journals, publishers, subject/research areas, and authors. The objective of the scientometric indicators was to examine the evolution of biolinguistics using CiteSpace and VOSviewer to explore indicators such as citation, co-citation, and co-occurrence.

The following is a summary of the key findings of this study based on the bibliometric analysis. (1) The last two decades have witnessed a remarkable increase in the production of knowledge in biolinguistics, as evidenced by the fact that of the 8285 biolinguistics publications published between 1935 and 2022, 7797 were published between 2000 and 2022. (2) The United States, United Kingdom, Europe, Australia, Japan, and China produce the most biolinguistics-related knowledge. (3) The top higher education institutions producing knowledge are located in the United Kingdom, Germany, the Netherlands, and other top 10 ranked nations. (4) The leading biolinguistics journals publish research from a variety of disciplines, including psychology, linguistics, cognitive sciences, neuroscience, and genetics. (5) Although Springer and Elsevier were the leading publishers of research in biolinguistics, all other publishers also publish research in this field. (6) Biolinguistics is an interdisciplinary field, and the publications we analysed were dispersed across various research/subject areas, such as the social sciences, arts and humanities, psychology, linguistics, computer science, and cognitive science. (7) Among the top authors producing more research in biolinguistics were Benitez-Burraco [104,167], Kirby and Christiansen [106], Kirby [168], and Boeckx [86].

There are at least five interpretations for these findings. First, the increase in the production of biolinguistics-related knowledge over the past two decades may be attributable to the emergence of linguistic theories, evolutionary developmental biolinguistics, and technologically advanced tools and software to track the development of biolinguistics. Another reason may be the increase in the number of people diagnosed with speech and language disorders or other disabilities manifesting as speech disorders. This in some way encourages more researchers to investigate the genetics of speech and language disorders in an effort to develop more effective preventive and therapeutic measures. The prevailing examples of their contributions include: Kirby [114], Chomsky [169,170], Pinker [107,171], and Arbib [143].

Second, the bibliometrics and scientometrics analysis of the biolinguistics discipline revealed that it is a rich, inter-disciplinary field with extensive ties to biology, language, psychology, and development. The contribution of biolinguistics to the study of language development and language evolution is growing rapidly day by day, particularly with the current development of technology that is linked to predetermined hypotheses of how language is processed and how the mind is endowed with soft-wired and innate processing mechanisms.

Thirdly, researchers in the field of biolinguistics should find our findings identifying the most sought-after keywords in biolinguistics useful. Human experiment, computer simulation, formal language, mathematical language, learning system, biological evolution, young adults, game theory, and computational linguistics were among these key terms. These keywords demonstrate the interdisciplinary nature of biolinguistics, as some are related to computer science (e.g., mathematical model), biology, and other disciplines (e.g., biological evolution). In a second group of keywords were cultural evolution, arbitrariness, gene, grammar, emergence and expansion, natural language, (language) faculty, and iconicity. Again, this pattern contains a greater number of linguistic-related and biology-related keywords. Together, these components constitute biolinguistics or the biology of language.

Fourth, since a large amount of data (8285 publications) was analysed in this study. There were no prior assumptions about the patterns or relationships among these data, other than the fact that they all use the term “biolinguistics” or other concepts that are equivalent (e.g., the biology of language). Using cluster analysis, however, we were able to classify these 8285 publications into 12 clusters representing 12 research patterns in biolinguistics. These patterns include included language evolution considering gestural communication [162], structural design of language, cultural evolution [151], neural networks [172], and language tools. Another set of patterns includes language evolution in relation to human language faculty in human language ready-brain [108], evolutionary biology in relation to road maps, and theoretical perspectives on language evolution. One more set of patterns is language evaluation in relation to human language evolution, cultural evolution, and molecular biology’s role in the study of the biological bases of human language [75].

Last but not least, what connects the previously generated clusters are either similar themes or authors’ approaches to controversial issues in biolinguistics, leading to more clustered research in a particular pattern. Again, in this study, we identified the central authors whose understanding of the connections between identified clusters was crucial. For instance, Kirby [105] was the central authors to establish connections related to the examination of language evolution and cultural evolution. Another example, Bickerton plays a vital role in understanding the connections between language evolution and structural design of language, and human language evolution and language evolution [173]. In addition to identifying the central author responsible for establishing the connection between these clusters in biolinguistics, we also identified the authors who are receiving more attention from other researchers and a rapid increase in citations. For instance, Hurford is a potential author of biolinguistics for his contribution in understanding language evolution through neural networks [160]. Another example is Chomsky who is intensively (co)-cited for his role in the emergence of the study of the evolution of language and the field of biolinguistics [48].

## 5. Practical Implications

Researchers should be cautious when interpreting the findings of scientometric studies [174] regardless of the popularity of this research method nowadays [175,176]. In the first instance, data should be retrieved from multiple sources and not limited to just one database unless it is well justified (e.g., in this study, we used Scopus, WOS, and Lens). A next step that should be taken is the use of different tools for the analysis to allow the inclusion of different scientometric indicators (e.g., in this study CiteSpace and VOSviewer were used).

## 6. Theoretical Implications

The challenge in biolinguistics research lies in providing concrete evidence concerning the biological basis of language. As far as reasoning human faculty as an innate human trait is concerned, the evidence and theories available are convincing yet arguable. Moreover, when examining human language using neuroscience and biolinguistics together, speech-language disorders of all kinds are also indicative of biological bases for human language. Current evidence is limited when it comes to identifying the origins or genetic basis of human language. A stronger evidence base and further development of the field of biolinguistics should be achieved through the integration of biolinguistics, neurolinguistics, psycholinguistics, and cognitive sciences. Another theoretical implication of this study is the need to promote interdisciplinary linguistics such as biolinguistics more. As opposed to focusing on the theoretical aspects of language, interdisciplinary fields of study are more in line with the requirements of contemporary challenges and can produce students with greater practical knowledge. Universities everywhere should re-evaluate their linguistics programmes in order to shift from traditional and outdated curriculum plans to those of the 21st century. There is a need to encourage more interdisciplinary research as opposed to unidirectional education and research.

## 7. Limitations and Future Directions

Future research could address a number of the limitations of this study. For instance, because we wanted to concentrate our research on the use of the terms “biolinguistics”, “biology of language”, and “language evolution”, we restricted our search strings to these terms. The incorporation of more specific concepts into biolinguistics would be the next step (e.g., genetics of speech, genetics of language, etc.). Although our cluster analysis helped identify patterns among more than 8000 biolinguistics publications, these clusters were not examined in detail. Future research could examine these clusters in depth in search of patterns of divergence and convergence. Another limitation is data analysis, specifically data merging. Although we intended to demonstrate how bibliometric and scientometric indicators could vary based on the database used and cautioned researchers against making broad assumptions based on a single database, merging the data would have improved the presentation of the results. We presented numerous figures and tables, which could have been condensed if the data had been merged and it had been possible to conduct the analysis using merged data.

## 8. Conclusions

The findings of this study provide evidence that biolinguistics is an independent field that also integrates with linguistics, biology, cognitive sciences, neuroscience, and anthropology. The analysis of 8285 biolinguistics publications between 1935 and 2022 revealed that 7797 were published between 2000 and 2022, indicating a significant increase in the production of knowledge in the field over the past two decades. In addition to identifying the leading biolinguistics-producing regions (e.g., the United States and the United Kingdom), we also identified the leading higher education institutions, journals, publishers, and authors. Importantly, we grouped the 8285 biolinguistics documents into clusters that represent the most popular search themes and topics in the field. This included the evolution of language taking into account gestural communication, linguistic structure, cultural evolution, neural networks, and language tools. Language evolution in relation to human language faculty in human language ready-brain, evolutionary biology in relation to road maps, and theoretical perspectives on language evolution comprise a second set of patterns. Language evaluation in relation to Grice’s host, human language evolution, cultural evolution, and molecular biology’s role in the study of biological bases of human language is a further set of patterns.

## Figures and Tables

**Figure 1 children-09-01300-f001:**
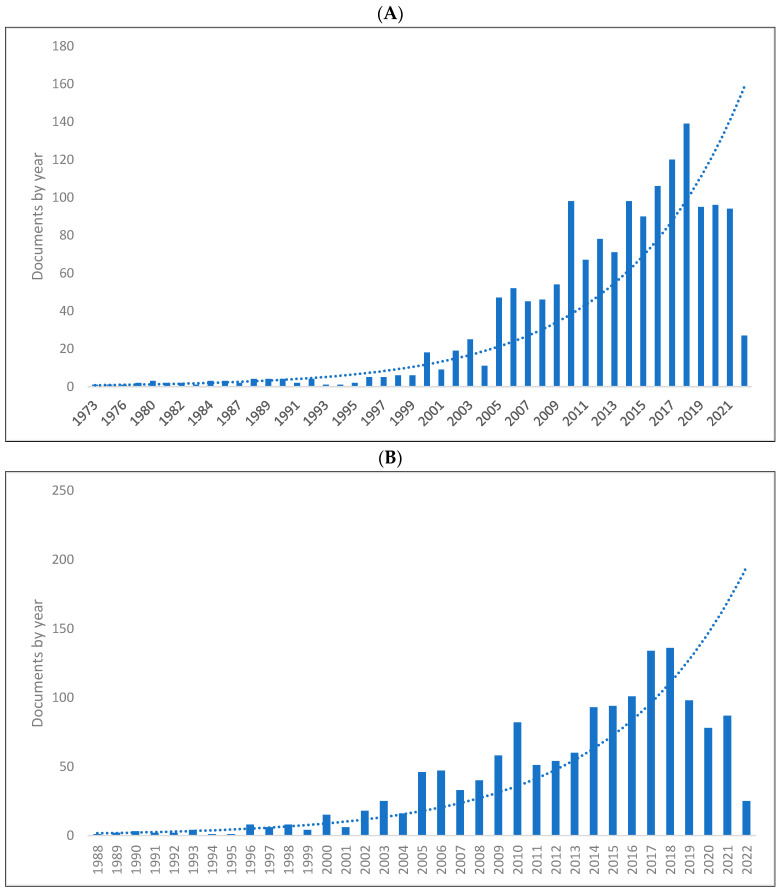
Knowledge production size in biolinguistics by year. (**A**) (Scopus), (**B**) (WOS), (**C**) (Lens).

**Figure 2 children-09-01300-f002:**
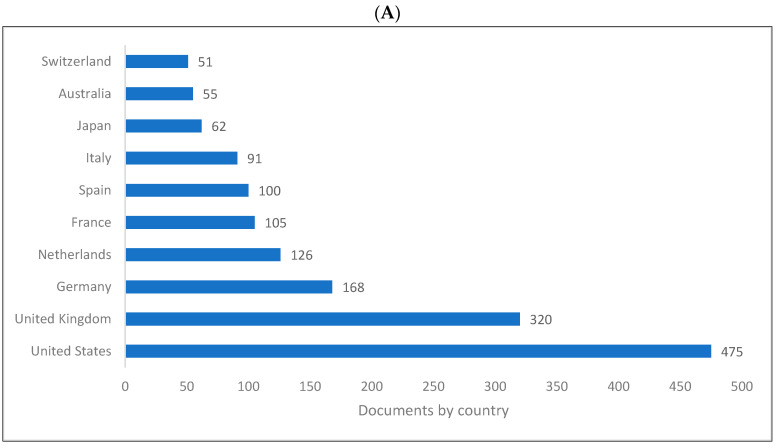
Knowledge production size in biolinguistics by country. (**A**) (Scopus), (**B**) (WOS), (**C**) (Lens).

**Figure 3 children-09-01300-f003:**
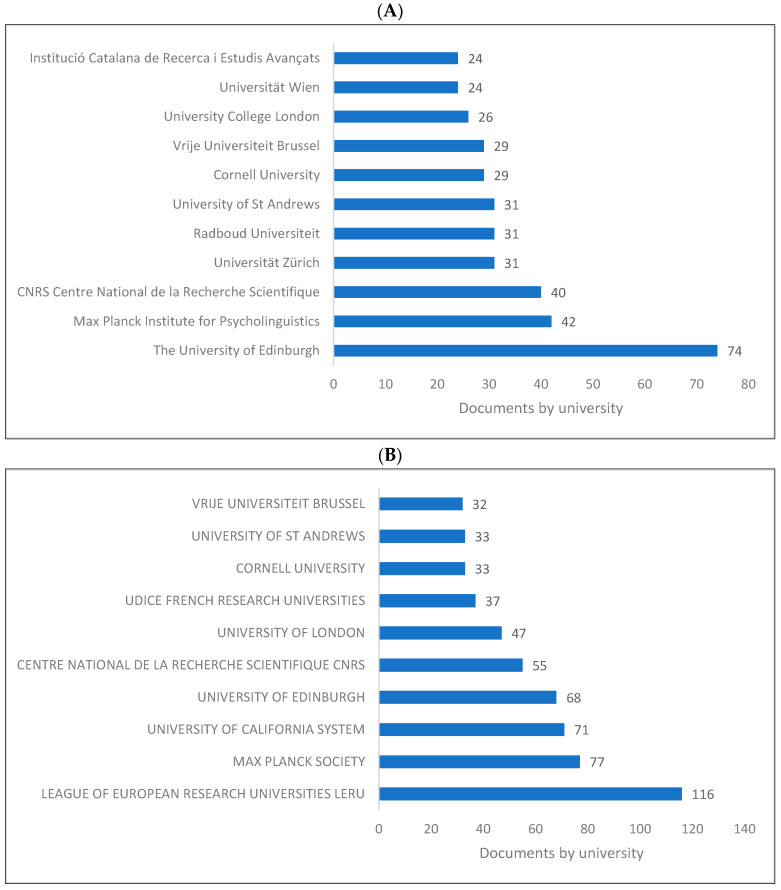
Knowledge production size in biolinguistics by university/research centre. (**A**) (Scopus), (**B**) (WOS), (**C**) (Lens).

**Figure 4 children-09-01300-f004:**
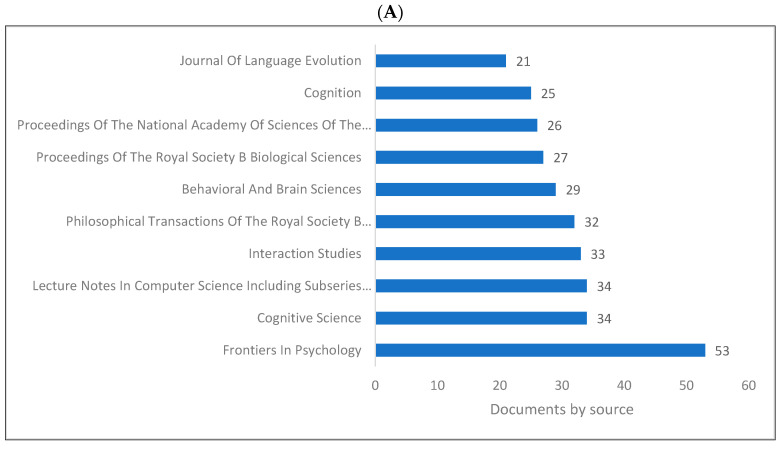
Knowledge production size in biolinguistics by journal. (**A**) (Scopus), (**B**) (WOS), (**C**) (Lens), (**D**) (Lens).

**Figure 5 children-09-01300-f005:**
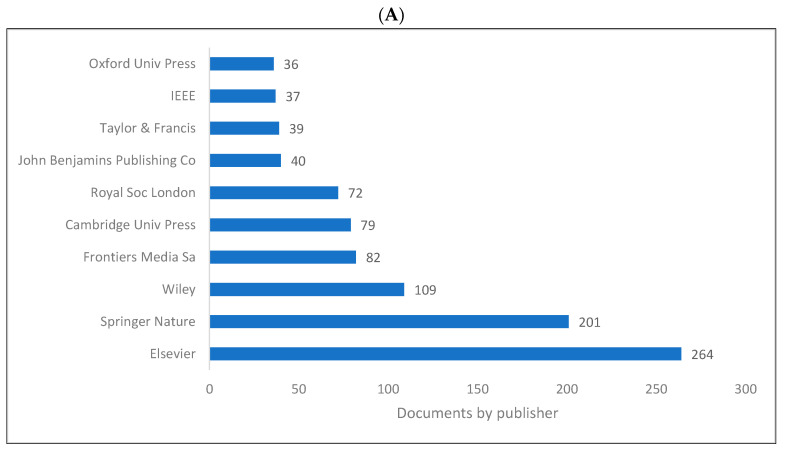
Knowledge production size in biolinguistics by publisher. (**A**) (WOS), (**B**) (Lens).

**Figure 6 children-09-01300-f006:**
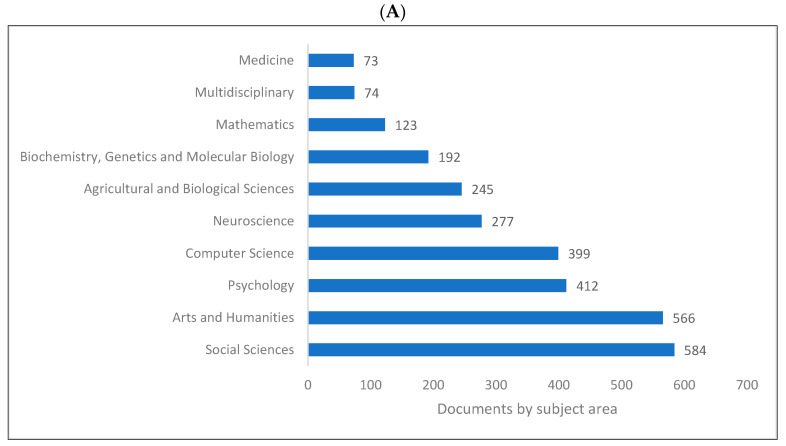
Knowledge production size in biolinguistics by research area. (**A**) (Scopus), (**B**) (WOS), (**C**) (Lens).

**Figure 7 children-09-01300-f007:**
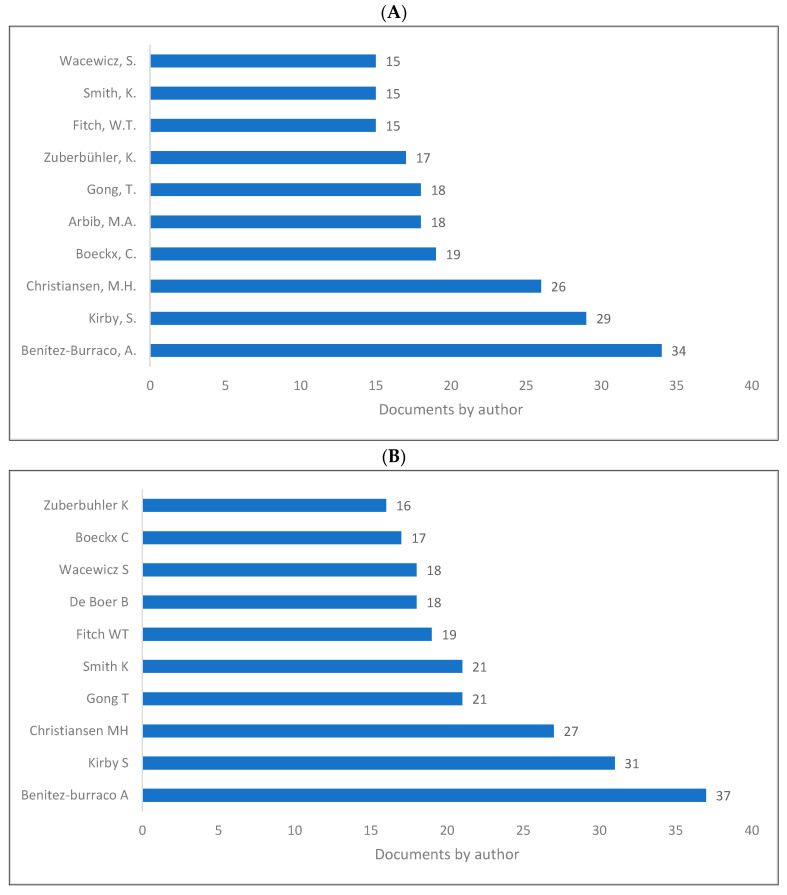
Knowledge production size in biolinguistics by author. (**A**) (Scopus), (**B**) (WOS), (**C**) (Lens).

**Figure 8 children-09-01300-f008:**
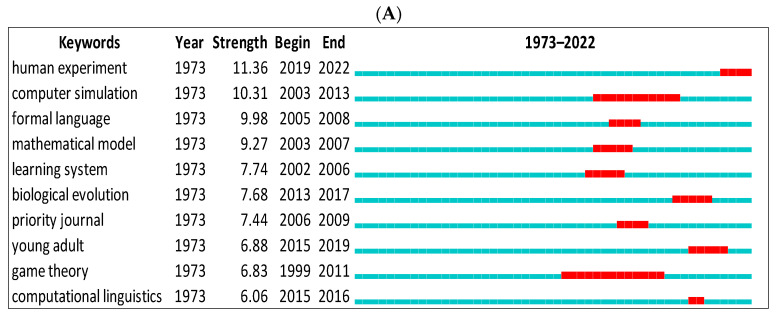
Top 10 keywords with the strongest citation bursts. (**A**) (Scopus), (**B**) (WOS).

**Figure 9 children-09-01300-f009:**
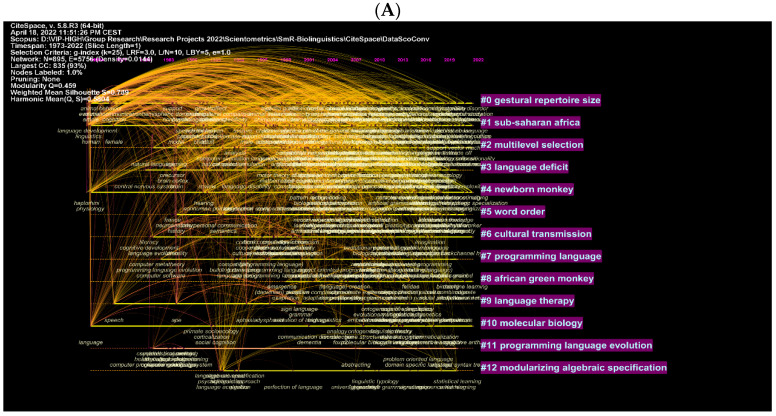
Top keywords, cited authors and clusters in biolinguistics. (**A**) (Scopus), (**B**) (WOS), (**C**) (Scopus), (**D**) (WOS).

**Figure 10 children-09-01300-f010:**
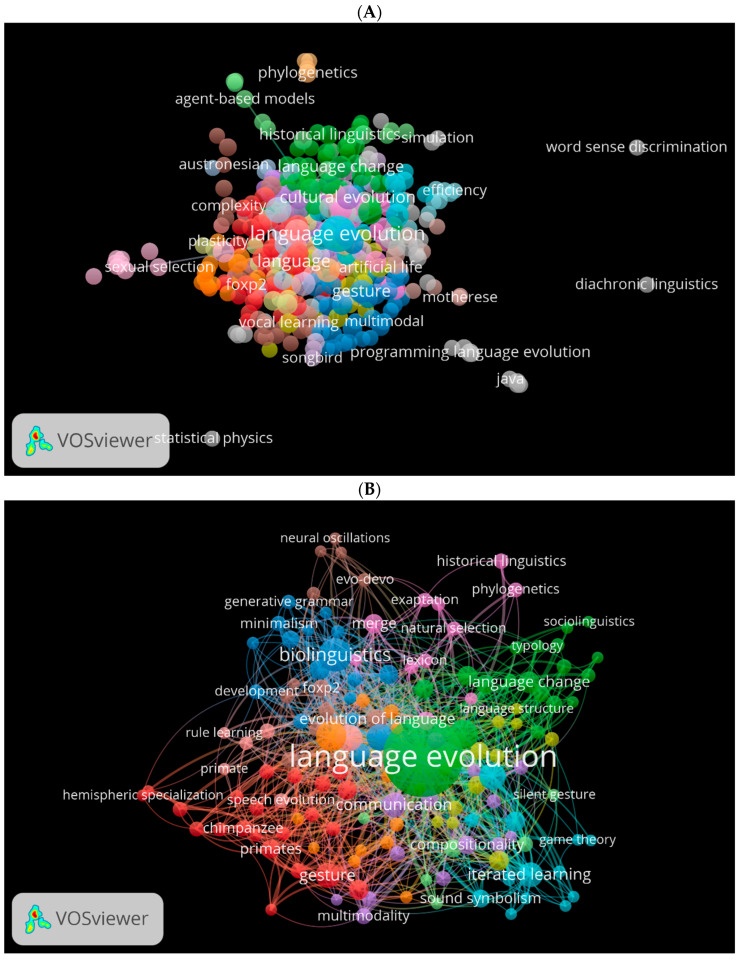
Cooccurrence by author keywords network visualisation in biolinguistics. (**A**) (Scopus), (**B**) (WOS), (**C**) (Lens).

**Figure 11 children-09-01300-f011:**
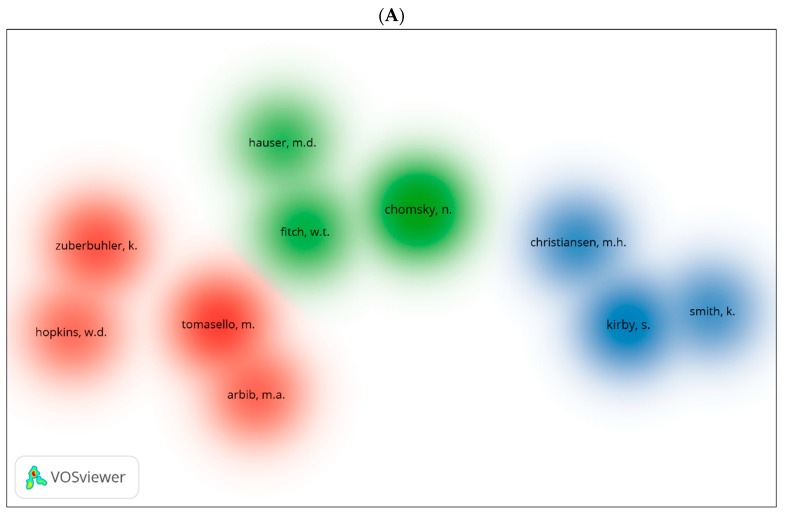
Co-citation by cited author density visualisation in biolinguistics. (**A**) (Scopus), (**B**) (WOS), (**C**) (Lens) citation by author density visualisation.

**Figure 12 children-09-01300-f012:**
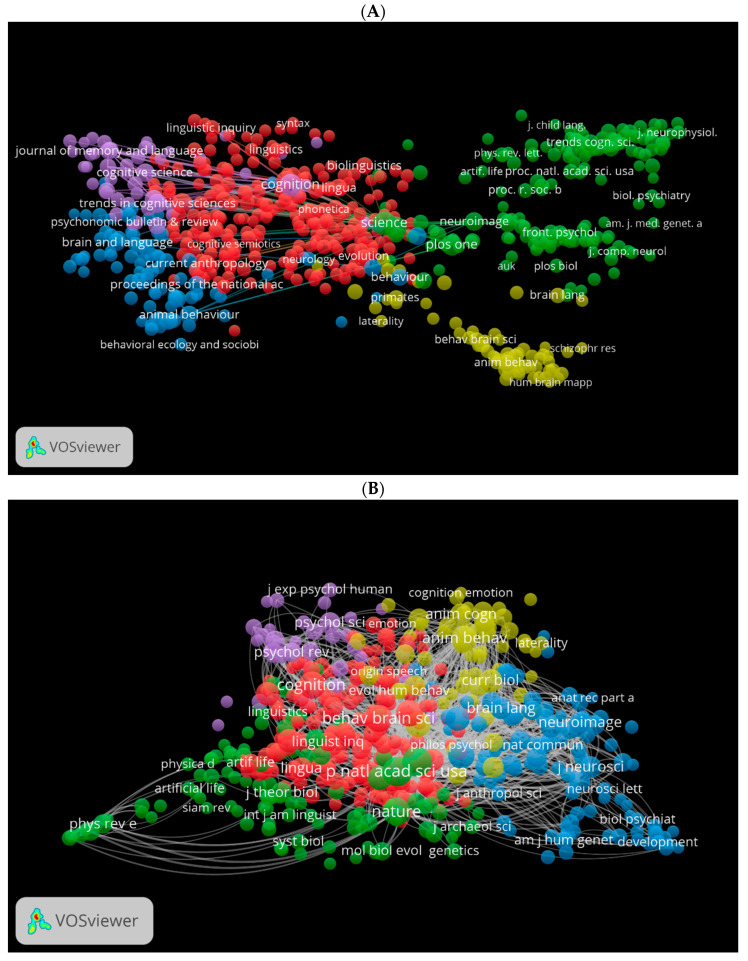
Co-citation by source network visualisation in biolinguistics. (**A**) (Scopus), (**B**) (WOS), (**C**) (Lens) citation by source network visualisation.

**Figure 13 children-09-01300-f013:**
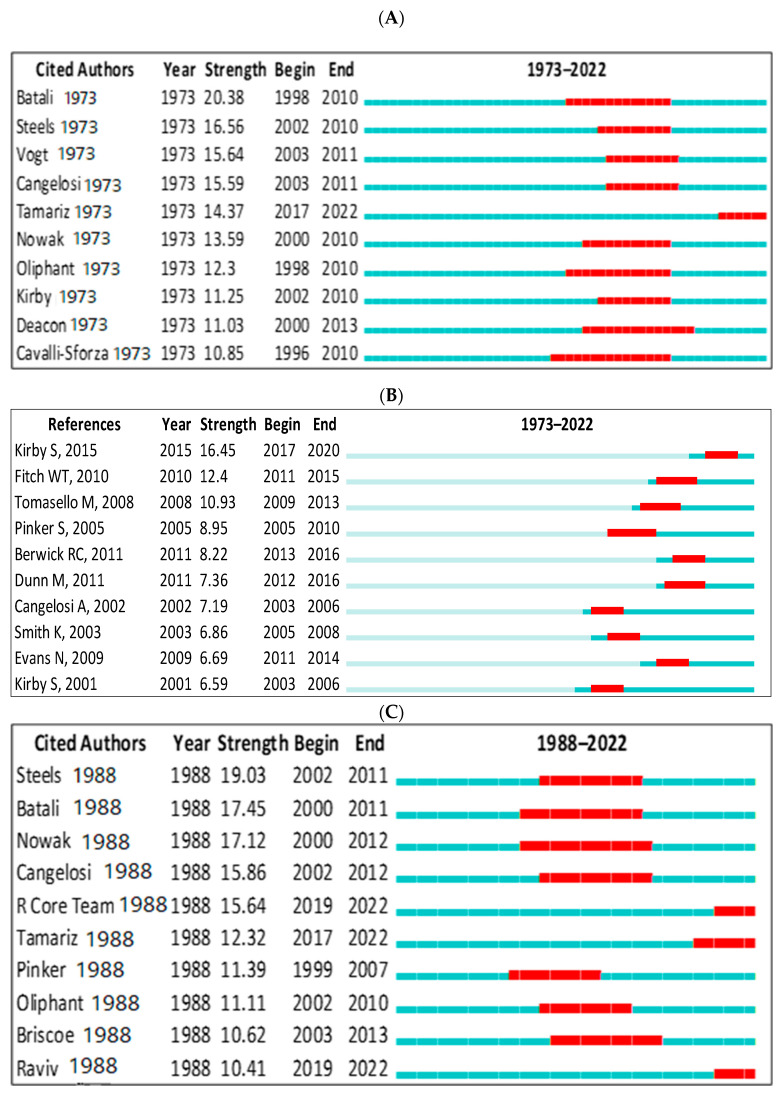
Top 10 cited authors and references with the strongest citation bursts. (**A**) (Scopus: cited authors), (**B**) (references), (**C**) (WOS: cited authors), (**D**) (WOS: references) [39,105,114,120,131,132,134,135,136,140,141,145,147,148,149,150,151,152,153,154,155,156,157].

**Table 1 children-09-01300-t001:** A List of bibliometric and scientometric indicators based on [94].

Element	Definition/Specification/Retrieved Data	Database/Software			
Indicator		Scopus	WOS		Lens
Bibliometric					
Year	Production size by year	√	√		√
Country	Top countries publishing in the field	√	√		√
University	Top universities, research centres, etc.	√	√		√
Source	Top journals, book series, etc.	√	√		√
Publisher	Top publishers	×	√		√
Subject area	Top fields associated with the field	√	√		√
Author	Top authors publishing in the field	√	√		√
Citation	Top cited documents	√	√		√
Scientometric		CiteSpace		VOSviewer	
Betweenness centrality	A path between nodes and is achieved when located between two nodes [101]	√		×	
Burst detection	Determines the frequency of a certain event in certain period (e.g., the frequent citation of a certain reference during a period of time) [102]	√		×	
Co-citation	When two references are cited by a third reference [103]. CiteSpace provides document co-citation network for references, and author co-citation network for authors.In VOSviewer, co-citation defined as “the relatedness of items is determined based on the number of times they are cited together” [100] (p. 5). Units of analysis include cited authors, references, or sources.	√		√	
Silhouette	Used in cluster analysis to measure consistency of each cluster with its related nodes [99]	√		×	
Sigma	To measure strength of a node in terms of betweenness centrality citation burst [99]	√		×	
Clusters	“We can probably eyeball the visualized network and identify some prominent groupings” [99] (p. 23).	√		√	
Citation	“The relatedness of items is determined based on the number of times they cite each other” [100] (p. 5). Units of analysis include documents, sources, authors, organizations, or countries.	√		√	
Keywords	CiteSpace provides co-occurring author keywords and keywords plus.In VOSviewer, co-occurrence analysis is defined as “the relatedness of items is determined based on the number of documents in which they occur together” [100] (p. 5). Units of analysis include author keywords, all keywords, or keywords plus.	√		√	

**Table 2 children-09-01300-t002:** Used search strings for biolinguistics in Scopus, WOS and Lens.

**Scopus** (TITLE-ABS-KEY ({biolinguistics}) OR TITLE-ABS-KEY (“biology of language”) OR TITLE-ABS-KEY (“language evolution”)) AND (LIMIT-TO(DOCTYPE, “ar”) OR LIMIT-TO (DOCTYPE, “cp”) OR LIMIT-TO (DOCTYPE, “re”) OR LIMIT-TO (DOCTYPE, “ch”) OR LIMIT-TO (DOCTYPE, “bk”)) Monday, 18 April, 2022, 1570 document results, 1973–2022
**WOS** “biolinguistics” (Topic) or “biology of language” (Topic) or “language evolution” (Topic) and Articles or Proceedings Papers or Editorial Materials or Review Articles or Book Chapters or Book Reviews or Early Access or Books (Document Types) Monday, 18 April, 2022, 1440 results, 1988–2022
**Lens** (Title: (AND (biolinguistics AND)) OR (Abstract: (AND (biolinguistics AND)) OR Full Text: (AND (biolinguistics AND)))) OR ((Title: (AND (“biology of language” AND)) OR (Abstract: (AND (“biology of language” AND)) OR Full Text: (AND (“biology of language” AND)))) OR (Title: (AND (“language evolution” AND)) OR (Abstract: (AND (“language evolution” AND)) OR Full Text: (AND (“language evolution” AND))))) **Filters**: Stemming = Disabled Publication Type = (journal article, unknown, book chapter, book, conference proceedings article, dissertation, preprint, conference proceedings) Author Display Name = (excl Chris J Myers, excl Michael Hucka) Monday, 18 April, 2022, Scholarly Works (5275), 1935–2022

**Table 3 children-09-01300-t003:** Top cited documents of biolinguistics based on citation reports from Scopus, WOS and Lens.

No.	Source Title	Citation	Citations by Database		
			Scopus	WOS	Lens
1	Language, usage and cognition	[109]	1089	×	780
2	Natural language and natural selection	[110]	949	×	1825
3	From monkey-like action recognition to human language: An evolutionary framework for neurolinguistics	[108]	656	574	×
4	Empathy and the Somatotopic Auditory Mirror System in Humans	[111]	537	503	×
5	Language as shaped by the brain	[112]	483	449	×
6	Language as a complex adaptive system: Position paper	[113]	415	411	×
7	Cumulative cultural evolution in the laboratory: An experimental approach to the origins of structure in human language	[114]	407	382	×
8	From mouth to hand: Gesture, speech, and the evolution of right-handedness	[115]	398	373	×
9	Least effort and the origins of scaling in human language	[116]	338	301	×
10	The evolution of language	[117]	315	×	844
11	A molecular genetic framework for testing hypotheses about language evolution	[118]	×	500	×
12	An introduction to evolutionary musicology	[119]	×	287	×
13	The evolution of the language faculty: Clarifications and implications	[120]	×	299	×
14	Cognition, evolution, and behavior	[121]	×	×	1395
15	Three Factors in Language Design	[122]	×	×	1314
16	The Evolution of Communication	[123]	×	×	1211
17	The small world of human language	[124]	×	×	855
18	The Biology and Evolution of Language	[125]	×	×	813
19	The Ecology of Language Evolution	[126]	×	×	797
20	Reactome knowledgebase of human biological pathways and processes.	[127]	×	×	780

**Table 4 children-09-01300-t004:** Summary of the largest clusters of biolinguistics using scientometric indicators.

Cluster ID	Size	Silhouette	Label (LSI)	Label (LLR)	Label (MI)	Average Year
Scopus						
0	210	0.728	Language evolution	Gestural communication (556.54, 1.0 × 10^−4^)	Year (1.53)	2010
1	191	0.594	Language evolution	Structural design (208.62, 1.0 × 10^−4^)	Year (1.86)	2006
2	183	0.722	Language evolution	Cultural evolution (755.77, 1.0 × 10^−4^)	Year (3.39)	2012
3	70	0.822	Language evolution	Neural network (282.82, 1.0 × 10^−4^)	Year (0.56)	2005
4	55	0.944	Language evolution	Sprachwerkzeuge al (language tools) (260.18, 1.0 × 10^−4^)	Linguistic perspective (0.54)	1984
5	53	0.91	Language evolution	Human language faculty (295.06, 1.0 × 10^−4^)	Human language-ready brain (0.17)	2000
6	38	0.926	Language evolution	Evolutionary biology (332.38, 1.0 × 10^−4^)	Road map (0.24)	2001
7	36	0.984	Theoretical perspective	Theoretical perspective (67.79, 1.0 × 10^−4^)	Language evolution (0.12)	1984
WOS						
0	171	0.791	Language evolution	Exorcising Grice’s ghost (305.11, 1.0 × 10^−4^)	Role (0.92)	2011
1	158	0.705	Language evolution	Human language evolution (269.87, 1.0 × 10^−4^)	Role (1.82)	2003
2	141	0.821	Language evolution	Language evolution (620.39, 1.0 × 10^−4^)	Role (1.94)	2013
3	132	0.746	Language evolution	Cultural variation (316.76, 1.0 × 10^−4^)	Role (2.02)	2005
4	120	0.745	Language evolution	Molecular biology (466.71, 1.0 × 10^−4^)	Role (1.13)	2013

**Table 5 children-09-01300-t005:** Citation counts of biolinguistics by references using scientometric indicators.

WoS			Scopus		
Citation	Reference	Cluster ID	Citation	Reference	Cluster ID
316	Hauser [132]	1	378	Chomsky [130]	4
308	Chomsky [133]	1	340	Tomasello [131]	0
303	Fitch [134]	1	298	Pinker [135]	1
294	Pinker [135]	1	280	Kirby [105]	2
279	Tomasello [136]	1	277	Hauser [137]	1
274	Kirby [105]	2	276	Fitch [120]	1
242	Chomsky [138]	1	231	Bickerton [139]	1
209	Bickerton [37]	1	202	Steels [140]	3
188	Christiansen [141]	2	194	Christiansen [142]	2
187	Arbib [143]	1	179	Jackendoff [144]	1

**Table 6 children-09-01300-t006:** Citation bursts of biolinguistics by references using scientometric indicators.

WoS			Scopus		
Burst	Reference	Cluster ID	Burst	Reference	Cluster ID
19.03	Steels [113]	3	20.38	Batali [145]	3
17.45	Batali [146]	3	16.56	Steels [140]	3
17.12	Nowak [147]	3	15.64	Vogt [148]	3
15.86	Cangelosi [149]	3	15.59	Cangelosi [149]	3
15.64	R Core Team [150]	2	14.37	Tamariz [151]	2
12.32	Tamariz [151]	2	13.59	Nowak [147]	6
11.39	Pinker [135]	1	12.3	Oliphant [152]	3
11.11	Oliphant [153]	3	11.25	Kirby [105]	2
10.62	Briscoe [154]	3	11.03	Deacon [155]	1
10.41	Raviv [156]	2	10.85	Cavalli-Sforza [157]	6

**Table 7 children-09-01300-t007:** Centrality of biolinguistics by references using scientometric indicators.

WoS			Scopus		
Centrality	Reference	Cluster ID	Centrality	Reference	Cluster ID
103	Kirby [105]	2	108	Donald [158]	0
99	Bickerton [37]	1	103	Bates [159]	0
96	Arbib [143]	1	102	Hurford [160]	1
95	Cheney [161]	0	101	Cheney [161]	0
93	Corballis [162]	1	98	Kirby [105]	2
91	Tomasello [136]	1	94	Lieberman [163]	1
90	Hurford [160]	3	91	Chomsky [130]	4
82	Fitch [134]	1	89	Darwin [164]	1
81	Rizzolatti [165]	0	85	Corballis [166]	0
81	Chomsky [15]	1	84	Bickerton [139]	1

**Table 8 children-09-01300-t008:** Sigma indicator of biolinguistics by references using scientometric indicators.

WoS			Scopus		
Sigma	Reference	Cluster ID	Sigma	Reference	Cluster ID
0	Kirby [105]	2	0	Donald [158]	0
0	Bickerton [37]	1	0	Bates [159]	0
0	Arbib [143]	1	0	Hurford [160]	1
0	Cheney [161]	0	0	Cheney [161]	0
0	Corballis [162]	1	0	Kirby [105]	2
0	Tomasello [136]	1	0	Lieberman [163]	1
0	Hurford [160]	3	0	Chomsky [130]	4
0	Fitch [134]	1	0	Darwin [164]	1
0	Rizzolatti [165]	0	0	Corballis [166]	0
0	Chomsky [15]	1	0	Bickerton [139]	1

## Data Availability

The data presented in this study are available on request from the first author.

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
