# Peer review of "Biolinguistics: A Scientometric Analysis of Research on (Children’s) Molecular Genetics of Speech and Language (Disorders)"

_children, 2022, doi:10.3390/children9091300_

Round 1

Reviewer 1 Report

This is an eyeopening analysis of what has been published on biolinguistics. I just have one question for the authors: why were the search results from the three databases analysed separately? Can these results be merged for analysis? I think that the overall trends across the three databases are similar. Was an analysis of the combined search results eve considered? 

Reviewer 2 Report

The results section makes for a long read. It is very descriptive and any analysis is left for a discussion, which is also fairly descriptive and makes no attempt to paint a bigger picture for the reader. Unfortunately, for me as a reader, this makes it not very interesting.

I would urge the authors to ask why they are actually carrying out this study. There is no critical engagement with this, and it bothers me that it is treated as a type of theory rather than a methodology in the paper. What is the actual question that they want to know the answer to and why is their method the best one to find this out?

The study is pretty meticulous and has rendered a lot of data, but I don’t really think this is enough.

Please strengthen the article in its theoretical background, add more support data, and give conclusions less simplistic.

Round 2

Reviewer 2 Report

N/a